# Look, Compare and Draw: Differential Query Transformer for Automatic Oil Painting

## Abstract

This work introduces a new approach to automatic oil painting that emphasizes the creation of dynamic and expressive brushstrokes. A pivotal challenge lies in mitigating the duplicate and common-place strokes, which often lead to less aesthetic outcomes. Inspired from the human painting process, *i.e.*, observing, comparing, and drawing, we incorporate differential image analysis into a neural oil painting model, allowing the model to effectively concentrate on the incremental impact of successive brushstrokes. To operationalize this concept, we propose the Differential Query Transformer (DQ-Transformer), a new architecture that leverages differentially derived image representations enriched with positional encoding to guide the stroke prediction process. This integration enables the model to maintain heightened sensitivity to local details, resulting in more refined and nuanced stroke generation. Furthermore, we incorporate adversarial training into our framework, enhancing the accuracy of stroke prediction and thereby improving the overall realism and fidelity of the synthesized paintings. Extensive qualitative evaluations, complemented by a controlled user study, validate that our DQ-Transformer surpasses existing methods in both visual realism and artistic authenticity, typically achieving these results with fewer strokes. The stroke-by-stroke painting animations are available on our anonymous website: `https://differential-query-painter.github.io/DQ-painter/`.

## 1 Introduction

Painting is a common form of human artistic expression, but it requires a certain level of technical skill. Computer-aided art enables people without professional drawing skills to create their own artistic works. Neural oil painting, which is based on stroke simulation, is one of the current approaches for transforming natural images into artistic renditions (Hertzmann, 2003; Singh et al., 2021; Liang et al., 2022; Wang et al., 2023; 2024). It aims to guide machines in progressively generating images by emulating authentic oil painting brushstrokes, from coarse to fine, on a digital canvas, thereby imparting to the images the characteristic texture of oil paintings.

Traditional stroke-based rendering methods typically rely on step-wise greedy search and heuristic optimization, which often lead to low efficiency (Haeberli, 1990; Litwinowicz, 1997; Tong et al., 2022). In recent years, deep learning-based methods have gained traction, employing a variety of strategies such as reinforcement learning (Huang et al., 2019; Singh et al., 2021; Wang et al., 2024; Hu et al., 2023), feed-forward neural networks (Liu et al., 2021), and optimization-based approaches (Zou et al., 2021; Kotovenko et al., 2021). While these methods have validated promising painting results, challenges in achieving higher efficiency and effectiveness in practical applications persist. For example, Hu et al. (2023) develop a reinforcement learning-based agent trained on real images (*e.g.*, ImageNet (Deng et al., 2009)) to dynamically determine the painting sequence, but it struggles with generalization, and becomes unstable when faced with unseen images. Similarly, Zou et al. (2021) introduce a stroke optimization method that achieves high-quality results but requires extremely long inference times. On the other hand, Liu et al. (2021) adopt a feed-forward approach using synthesized stroke images to efficiently predict sets of strokes. However, this method often produces coarse strokes and particularly fails to capture fine details at the canvas boundaries.

Despite varying learning strategies within specific models, the prevailing works on neural oil painting all adhere to the iterative learning paradigm, that is, predicting the subsequent brushstroke based

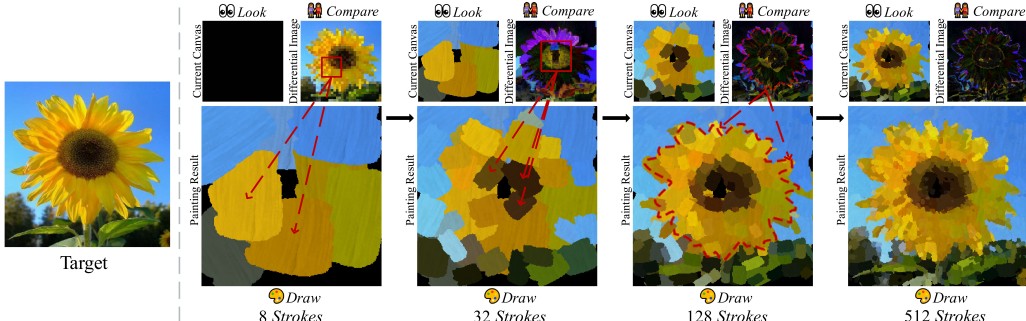

Figure 1: Differential image-guided inference process. We present four intermediate stages of oil painting according to a real target image (left). Each stage is illustrated with a diagram, where the top-left corner shows the current canvas, the top-right corner displays the corresponding differential image for that stage, and the bottom part presents the painting result inferred by our model. We observe that since we explicitly compare the content in the differential images during training, our model tends to add strokes in areas where discrepancies are more pronounced, thereby progressively reducing the discrepancy content within the differential images.

on the current one. In line with this learning paradigm, existing methodologies employ a rather direct approach by generating the forthcoming brushstroke directly using the existing stroke as input. We contend that this predictive approach suffers from the absence of an intermediate guidance from the current stroke to the next, which becomes particularly challenging when there is a significant divergence between the paintings in the early steps of prediction. Conversely, in the human painting process, artists frequently observe and compare the difference between their current work and the target painting before deciding on the subsequent brushwork. Motivated by this procedure, we propose the incorporation of image discrepancy as a form of intermediate guidance to address the neural oil painting problem, aiming to bridge the gap between the current iteration and the ultimate artistic vision, thereby enhancing the fidelity and effectiveness of the neural painting process.

Based on the above considerations, we propose a differential image-guided painter framework: the Differential Query Transformer (DQ-Transformer). The DQ-Transformer learns differential image features between the current canvas and the target image, focusing on the discrepancies between the images, thereby enabling more accurate stroke predictions. In particular, we employ local encoders comprised of convolutional neural networks to learn three position-aware image features separately: the current canvas, the target image, and the differential image between these two. The differential image features are then transformed into query tokens, which are used as queries to the DQ-Transformer to decode the stroke parameters. The final painting result is obtained by rendering these decoded strokes onto the canvas. We first minimize the $L_1$ distance between the target image and the rendered image, as well as the $L_1$ distance between the predicted strokes and the ground-truth strokes. Furthermore, we train the DQ-Transformer with a WGAN-based discriminator (Ganin et al., 2018b), as optimizing only the $L_1$ distance loss leads to poor reconstruction accuracy. The discriminator is utilized during training to enhance the precision of predicted strokes, by treating the rendered images as fake samples and striving to penalize the generation of erroneous strokes.

The "look, compare and draw" painting process of our model is illustrated in Figure 1, where we present four intermediate stages of completing a real image with several strokes. It can be observed that our model evaluates the content of the differential image and introduces strokes precisely in areas exhibiting more significant disparities. This strategic addition of details progressively diminishes the discrepancies within the differential images, advancing toward a refined output. To prove that the oil paintings produced by our method are of high quality, we compare them with other state-of-the-art stroke-based oil painting methods. Qualitative comparisons indicate that our method can generate images with more authentic oil painting textures while maintaining the fidelity of the original images. We have conducted a Mean Opinion Score (MOS) test and invited volunteers to evaluate the quality of oil paintings created by the above methods. The paintings of our method attained the highest preference ratings from the users. The primary contributions of our work are:

- **Differential Image Analysis Integration:** We introduce a new painting pipeline that embeds differential image analysis within the neural oil painter framework. By focusing on the incremental changes wrought by successive brushstrokes, this simple and effective en-

hancement sharpens the attention to localized details, yielding a more intuitive and nuanced rendering process.

- **Differential Query Transformer Architecture:** Inspired by the spirit of human artists, *i.e.*, observing, comparing and drawing, we further introduce a Differential Query Transformer (DQ-Transformer) that explicitly leverages differential image features, enriched with positional encoding, which serve as queries to guide stroke prediction.

- **Superior Performance:** Both quantitative and qualitative experiments on three public datasets, *i.e.*, Landscapes, FFHQ, and Wiki Art, affirm that the proposed method achieves better pixel-level and perception-level reconstruction, as well as higher user preference across various painting themes. Furthermore, the proposed method is stroke-efficient, *i.e.*, it achieves competitive painting quality with fewer strokes.

## 2 RELATED WORK

Differing from pixel-based generative models (Zheng et al., 2019b; Ho et al., 2020; Meng et al., 2022; Zhang et al., 2023; Mou et al., 2023; Diao et al., 2023; Li et al., 2023; Chen et al., 2024a;b), automatic oil painting adopts the brushstrokes as the fundamental unit of creation. Traditional stroke-based methods (Haeberli, 1990; Litwinowicz, 1997; Turk & Banks, 1996) rely on a greedy stroke-searching strategy. Taking one step further, Im2Oil (Tong et al., 2022) combines adaptive sampling based on probability density maps, thereby producing remarkable painting results. These traditional search-based methods tend to have low search efficiency, particularly when dealing with problems that have a large search space, leading to lengthy runtimes.

Recently, deep learning-based methods have gained increasing popularity and various learning strategies have been explored to address stroke-based rendering problems. In particular, existing automatic oil painting methods, based on deep neural networks, can primarily be classified into three categories as follows: **(1) Optimization-based methods.** The optimization-based methods (Tang et al., 2017) aim to arrange the order of each stroke, improving the efficiency of drawing algorithms. Ashcroft et al. (2024) introduce a generative model for creating complex vector drawings and show its effectiveness in generating intricate anime line art. To better apply painting techniques to real-world images, Stylized Neural Painting (Zou et al., 2021) mimics the behavior of a vector graphics renderer, by treating stroke prediction as a parametric search process. Meanwhile, Parameterized Brushstrokes (Kotovenko et al., 2021) searches for various styles of parameterized brushstrokes to complete the painting. These methods can be optimized jointly with neural style transfer but suffer from long optimization times for each image. **(2) Feed-forward neural network-based methods.** The feed-forward neural network-based methods (Ha & Eck, 2018; Frans & Cheng, 2018; Zheng et al., 2019a) utilize basic neural architectures to predict the stroke sequences in paintings. In early research, Recurrent Neural Networks (RNNs) (Graves, 2013) decompose images into sequences, but the need for detailed, manually annotated datasets hinders progress. Aksan et al. (2020) propose CoSE, which decomposes sketches into a set of stroke collections to construct structured drawings. Furthermore, Liu et al. (2021) propose a Paint Transformer with a self-supervised pipeline, which accelerates the training stage and achieves better training stability. While their approach is computationally efficient and requires no additional annotations, the predicted strokes are coarse and tend to miss details at the boundaries of the canvas. **(3) Reinforcement learning-based methods.** The reinforcement learning-based methods (Ganin et al., 2018a; Zhou et al., 2018; Singh & Zheng, 2021; Singh et al., 2021; Wang et al., 2024) aim to learn the textures and styles of real-world images to improve the painting quality. As a seminal effort, Learning to Paint (Huang et al., 2019) employs a more complicated reinforcement learning model to paint complex real-world images with a water-color brush. Moreover, Compositional Neural Painter (Hu et al., 2023) incorporates object detection learning into the reinforcement learning model, dynamically segmenting and predicting stroke regions. Training a stable reinforcement learning agent is challenging due to the dynamic interactions among its components, as this process typically leads to instability.

Although the aforementioned methods achieve satisfactory results in rendering paintings, they suffer from issues such as boundary inconsistencies and struggle with more intricate images. We address these limitations by introducing a DQ-Transformer architecture that leverages differentially derived image representations, augmented with positional information, to guide informed stroke prediction. Our model is both sensitive to position and capable of producing higher-quality renderings.

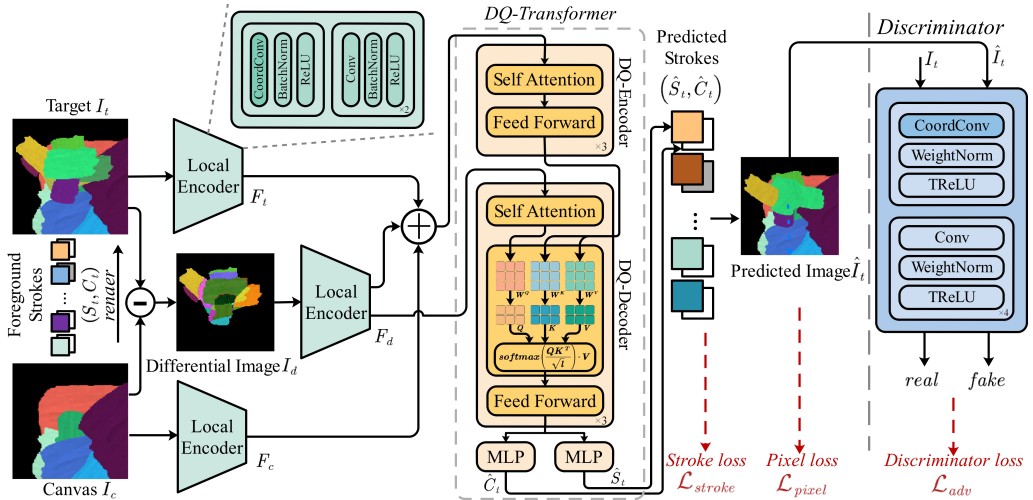

Figure 2: A brief overview of our painter framework. Given the canvas image $I_c$ and the target image $I_t$ generated by the renderer, we first obtain their differential image $I_d$ by simply subtracting one input from the other. Three local encoders comprised of convolutional neural networks are employed to extract image features $F_c, F_t$, and $F_d$ with positional information. DQ-Transformer has two components, *i.e.*, the DQ-encoder and the DQ-decoder. These visual features $F_c, F_t$ and $F_d$, are concatenated and then fed to the DQ-encoder to obtain the fused feature $F_{kv}$. Next, we transform the differential image features $F_d$ into query tokens to query the key and value pairs generated by the fused feature $F_{kv}$. Finally, the DQ-Transformer outputs a set of predicted strokes $\hat{S}_t$, each accompanied by its respective confidence $\hat{C}_t$. The predicted image $\hat{I}_t$ is generated by rendering these strokes onto the canvas. The discriminator operates by treating the target images $I_t$ as real samples and the predicted images $\hat{I}_t$ as fake samples.

## 3 METHODOLOGY

### 3.1 OVERVIEW

Neural painting simplifies the painting task into predicting a sequence of brush strokes. In this section, we provide a detailed introduction to the training process of our painter framework, as well as the inference process involved in creating artwork. A brief overview of our painter framework is illustrated in Figure 2. We employ a self-supervised pipeline in which the current canvas and target images are constructed using randomly synthesized strokes, eliminating the need for real images during training. In each training iteration, we first randomly sample two strokes sets: a background strokes set $S_b$ to generate the canvas $I_c$, and a foreground strokes set $S_t$ to create the target image $I_t$ based on $I_c$. Background strokes are rendered onto an empty canvas to establish the current canvas $I_c$. Subsequently, the foreground strokes are superimposed onto the current canvas to produce the target image $I_t$. Notably, the background strokes are coarser in granularity than the foreground strokes. This construction methodology mirrors the human artistic process, which evolves from broad outlines to detailed refinements. Furthermore, we construct a differential image between the target image and the current canvas, which subsequently serves as the query tokens for our DQ-Transformer. The differential operation approximates how the human visual system processes image information, emphasizing the incremental effects resulting from consecutive brushstrokes.

### 3.2 STROKE RENDERER

For stroke rendering, we adjust the properties of a real still brushstroke, *i.e.*, oil brushstroke (Zou et al., 2021), to create different stroke variants based on given parameters. The strokes parameters are $s = \{x, y, h, w, \theta, r, g, b\}$, where $(x, y)$ denotes the coordinates of the stroke center, $h, w$ represent the height and width of the stroke, $\theta$ is the rotation angle, and $(r, g, b)$ indicates the RGB values of the stroke. At each step $n$, the stroke renderer is employed to render the stroke parameters into a stroke image $R_n$ and a binary mask $M_n$, where $M_n$ is a single-channel alpha map of $R_n$. These stroke images are then sequentially added to the current canvas, potentially covering any previous

strokes if they exist. The iterative rendering process can be formulated as:

$$I_n = R_n \odot c_n M_n + I_{n-1} \odot (1 - c_n M_n), \tag{1}$$

where $c_n$ is the confidence of the stroke, indicating whether the stroke is valid. $\odot$ is the element-wise multiplication, while $I_{n-1}$ is the previous painting result. The entire rendering process is based on differentiable linear transformations and does not contain any trainable parameters.

### 3.3 PAINTER FRAMEWORK

The painter framework aims to reconstruct the target image $I_t$ using a sequence of predicted strokes. Given the current canvas $I_c \in \mathbb{R}^{3 \times P \times P}$ and the target image $I_t \in \mathbb{R}^{3 \times P \times P}$, where $P$ is the pre-defined patch size that acts as the basic unit for subsequent painting. Then the differential image is obtained by performing a pixel-wise subtraction: $I_d = I_t - I_c$. Our painter framework takes $I_c, I_t$, and $I_d$ as input and predicts a stroke set $\hat{S}_t$. The predicted image is generated by rendering these strokes onto the canvas, as described in Sec. 3.2.

**Local Encoder.** As shown in Figure 2, the painter framework first employs separate local encoders, comprised of convolutional neural networks, to individually extract their feature maps, denoted as $F_c, F_t, F_d \in \mathbb{R}^{3 \times \frac{P}{4} \times \frac{P}{4}}$. It is worth noting that traditional convolutional layers lack explicit positional encoding, and stacking them directly can lead to the loss of coordinate information. To address this issue, we substitute traditional convolutional layers with CoordConv (Liu et al., 2018), implementing it in the first layer of the convolutional network. CoordConv introduces additional channels to the input feature map, representing the coordinates of each feature pixel, thereby enabling the convolutional learning process to have a degree of awareness about the spatial positions. Then, $F_c, F_t$, and $F_d$, endowed with positional encoding, are concatenated and flattened as the input of DQ-Transformer.

**DQ-Transformer.** DQ-Transformer consists of two main parts: a DQ-Encoder and a DQ-Decoder. The DQ-Encoder block consists of a self-attention layer and a feed-forward layer, and it learns the concatenated features $\{F_c, F_t, F_d\}$ from the local encoders to produce the fused features $F_{kv}$. The DQ-Decoder block comprises a self-attention layer, a cross-attention layer, and a feed-forward layer. In the DQ-Decoder, the differential image features $F_d$ are transformed into query tokens. This transformation helps the model focus on local changes introduced by incremental strokes. The DQ-Decoder then considers the correspondences between the differential query tokens $F_d$ and the fused features $F_{kv}$ output by the DQ-encoder. The self-attention layer learns the relative attention and interactions among the various elements of differential query tokens. The cross-attention layer implements $CrossAttention\left(Q; K; V\right) = softmax\left(\frac{QK^T}{\sqrt{l}}\right) \cdot V$, and $l$ is the output dimension of key and query features, while

$$Q = W^Q F_d, K = W^K F_{kv}, V = W^V F_{kv}, \tag{2}$$

where $W^Q$, $W^K$, and $W^V$ are learnable weights that project $F_d$ to query, and map $F_{kv}$ to key and value, respectively. Finally, the differential query tokens are fed through two MLPs to predict stroke parameters $\hat{S}_t = \{\hat{s}_i\}_{i=1}^N$ and their corresponding confidences $\hat{C}_t = \{\hat{c}_i\}_{i=1}^N$ respectively. During the inference phase, we determine whether the predicted stroke is valid based on the sign of confidence $\hat{c}_i$. If $\hat{c}_i \geqslant 0$, we draw this stroke, otherwise, we skip it. We draw all predicted valid strokes onto the canvas, yielding the final painting $\hat{I}_t$.

### 3.4 TRAINING OBJECTIVE

**Pixel Loss.** The most direct goal of neural painting is to reconstruct the target image. Therefore, we minimize the $L_1$ distance between the predicted image $\hat{I}_t$ and the target image $I_t$:

$$\mathcal{L}_{pixel} = \lambda_p \left\| I_t - \hat{I}_t \right\|_1, \tag{3}$$

where $\lambda_p$ is a weight term.

**Stroke Loss.** Similarly, since the target image is rendered from the canvas image using the foreground strokes set, we can constrain the difference between ground-truth and prediction at the stroke

level. Considering the misordering of predicted strokes, we employ the Hungarian Algorithm (Kuhn, 1955) to perform an optimal bipartite matching between the set of predicted strokes and the set of ground-truth strokes. The stroke sets rearranged by the Hungarian Algorithm are represented as $u$ and $\hat{u}$ for target strokes and predicted strokes respectively. We define $L_1$ distance of stroke sets as:

$$\mathcal{D}_{L_1}^u = \|s_u - \hat{s}_u\|_1, \tag{4}$$

where $s_u$ and $\hat{s}_u$ denote parameters of strokes $u$ and $\hat{u}$ in $S_t$ and $\hat{S}_t$, respectively. Moreover, a rotational rectangular stroke with parameters $[x, y, h, w, \theta]$ can be viewed as a 2-D Gaussian distribution $\mathcal{N}(\mu, \tau)$ (Yang et al., 2021). Therefore, the Wasserstein distance between two strokes sets $\mathcal{N}(\mu_u, \tau_u)$ and $\mathcal{N}(\hat{\mu}_u, \hat{\tau}_u)$ is calculated by:

$$\mathcal{D}_W^u = \|\mu_u - \hat{\mu}_u\|_2^2 + Tr\left(\tau_u + \hat{\tau}_u - 2\left(\tau_u^{\frac{1}{2}}\hat{\tau}_u\tau_u^{\frac{1}{2}}\right)^{\frac{1}{2}}\right), \tag{5}$$

where $Tr(\cdot)$ denotes the trace of a matrix. Notably, with the predefined maximum stroke number $|S_t|$, we assign a confidence $c_i$ to each stroke $s_i$, implying that the number of valid strokes within each set can vary. The strokes in the prediction set $\hat{S}_t$ and the ground-truth set $S_t$ can be valid or empty. We utilize binary cross-entropy to match $c_u$ and $\hat{c}_u$:

$$\mathcal{D}_{bce}^u = -\left(\lambda_r \cdot c_u \cdot \log \sigma\left(\hat{c}_u\right) + (1 - c_u) \cdot \log\left(1 - \sigma\left(\hat{c}_u\right)\right)\right), \tag{6}$$

where $\lambda_r$ is a weight term for positive samples and $\sigma(\cdot)$ denotes the *sigmoid* function. Therefore, the total loss on the re-matched strokes can be formulated as:

$$\mathcal{D}_{match} = \frac{1}{|S_t|}\sum_{u=1}^{|S_t|}\left(c_u\left(\mathcal{D}_{L_1}^u + \lambda_W\mathcal{D}_W^u\right) + \mathcal{D}_{bce}^u\right), \tag{7}$$

where $\lambda_W$ is a weight term, and $|S_t|$ is the number of strokes. Finally, to encourage the model to reconstruct the target image using the minimum number of valid strokes, we impose an additional regularization on the confidence $\hat{C}_t$ of the predicted strokes. Therefore, the stroke loss is formulated as:

$$\mathcal{L}_{stroke} = \mathcal{D}_{match} + \lambda_c\frac{1}{|S_t|}\sum_{u=1}^{|S_t|}\|\hat{c}_u\|_1, \tag{8}$$

where $\lambda_c$ is a weight term for the confidence regularization.

**Adversarial Loss.** Treating our painting network as a generator, we observe that deep neural network-based painting networks achieve better reconstruction accuracy when coupled with a WGAN discriminator (Gulrajani et al., 2017). We have designed a simple discriminator network, which treats the generated images as fake samples, encouraging the model to predict strokes that make the painting more similar to the target image.

As shown in Figure 2, the discriminator consists of five blocks. Each block, except the first one, comprises Conv, WeightNorm, and TReLU layers. In the first block, we replace the Conv layer with a CoordConv layer. The training process employs a WGAN-GP loss function as:

$$\mathcal{L}_{adv} = Dis\left(\hat{I}_t\right) - Dis\left(I_t\right) + \lambda_{dis}\left(\left\|\nabla_{\tilde{I}_t}Dis\left(\tilde{I}_t\right)\right\|_2 - 1\right)^2, \tag{9}$$

where $Dis(\cdot)$ represents the discriminator score for a given sample. $\tilde{I}_t$ is a linear interpolation between real samples $I_t$ and fake samples $\hat{I}_t$. $\left\|\nabla_{\tilde{I}_t}Dis\left(\tilde{I}_t\right)\right\|_2$ is the $L2$ norm of the gradient of the discriminator on the interpolation point. $\lambda_{dis}$ is the hyperparameter for the gradient penalty.

**Overall loss.** Finally, our network is optimized by minimizing the pixel loss, the stroke loss, and the adversarial loss as:

$$\mathcal{L}_{total} = \mathcal{L}_{pixel} + \mathcal{L}_{stroke} + \gamma\mathcal{L}_{adv}, \tag{10}$$

where $\gamma = \frac{\|\mathcal{L}_{pixel}\|}{\|\mathcal{L}_{adv}\|}$ is an adaptive regularization factor for balancing.

Table 1: Quantitative comparison with competitive methods under pixel-level and perception-level reconstruction on unseen real-world datasets. Smaller values indicate better image reconstruction quality. All painting results are produced at a resolution of $512 \times 512$ pixels. The maximum number of valid strokes is set to 5000. *w/o $I_d$* denotes that we do not use the differential image, while *w/o* Reg ($\lambda_c = 0$) means the model without confidence regularization in Eq. 8, *w/o* CoordConv represents we solely employ conventional convolutional layers to extract image features, *w/o* Discriminator denotes that we train the model without the discriminator.

| Methods | Landscapes | | FFHQ | | Wiki Art | | Average | |
|---|---|---|---|---|---|---|---|---|
| | $\mathcal{L}_{pixel} \downarrow$ | $\mathcal{L}_{pcpt} \downarrow$ | $\mathcal{L}_{pixel} \downarrow$ | $\mathcal{L}_{pcpt} \downarrow$ | $\mathcal{L}_{pixel} \downarrow$ | $\mathcal{L}_{pcpt} \downarrow$ | $\mathcal{L}_{pixel} \downarrow$ | $\mathcal{L}_{pcpt} \downarrow$ |
| Stylized Neural Painting (Zou et al., 2021) | 0.068 | 0.939 | 0.057 | 1.044 | 0.064 | 0.996 | 0.063 | 0.993 |
| Paint Transformer (Liu et al., 2021) | 0.070 | 0.807 | 0.056 | 0.934 | 0.061 | 0.841 | 0.062 | 0.861 |
| Im2Oil (Tong et al., 2022) | 0.064 | 0.720 | 0.042 | 0.742 | 0.052 | 0.718 | 0.053 | 0.727 |
| Compositional Neural Painter (Hu et al., 2023) | 0.056 | 0.732 | **0.037** | 0.772 | 0.046 | 0.715 | **0.046** | 0.740 |
| *w/o $I_d$* | 0.078 | 0.833 | 0.064 | 0.975 | 0.066 | 0.868 | 0.069 | 0.892 |
| *w/o* Reg ($\lambda_c = 0$) | 0.064 | 0.476 | 0.048 | 0.791 | 0.055 | 0.736 | 0.056 | 0.668 |
| *w/o* CoordConv | 0.075 | 0.854 | 0.059 | 0.976 | 0.067 | 0.899 | 0.067 | 0.910 |
| *w/o* Discriminator | 0.059 | 0.735 | 0.047 | 0.713 | 0.051 | 0.770 | 0.052 | 0.739 |
| Ours | **0.054** | **0.579** | 0.039 | **0.631** | **0.045** | **0.593** | **0.046** | **0.601** |

## 3.5 PAINTING INFERENCE

To generate painting strokes that mimic human artists, we predict strokes in a coarse-to-fine manner during the inference process. Given a real image with shape $H \times W$, we first determine to process it from coarse to fine over $K$ scales and pad the image to a size of $P \times 2^K$, where $P$ is the predefined patch size. Both the target image and the current canvas are uniformly divided into multiple patches with a size of $P \times P$, which are then fed into our painting network for stroke prediction. At each scale, the initial canvas is the rendered image from the previous scale. In the $k$-th scale (where $0 \leqslant k \leqslant K$), there are $2^k \times 2^k$ patches. Each patch is processed by the painting network and then rendered in parallel. The painting result at each scale is achieved by combining the patches on the canvas. Moreover, after completing $K$ levels of painting, we further pad the target image with a size of $P$ and execute another an additional round of painting, which can help add more details to the current canvas image. Our coarse-to-fine painting process is illustrated in Figure 4 (see Appx. B for more details). It is noteworthy that there are no noticeable boundaries between patches in the final composite image. This is because our painting network is position-aware and does not disregard strokes at the canvas edges. It achieves this by adopting the differential image as a query and encoding image features through positional embeddings.

## 4 EXPERIMENT

### 4.1 IMPLEMENTATION DETAILS

Our model is trained exclusively using synthesized stroke images, without relying on any real-world datasets. We conduct evaluation on three distinct datasets: Landscapes (Chen et al., 2018), FFHQ (Karras et al., 2019), and Wiki Art (Phillips & Mackintosh, 2011). For each dataset, we randomly select 100 images as test samples. We set patch size $P$ as 32 and the maximum number of brushstrokes $|S_t|$ in one patch as 8. During training, parameters for target strokes are randomly generated from a uniform distribution. We sequentially render these strokes, and if a stroke covers more than 75% of the area of the preceding stroke, its confidence is set to 0 to ensure that the rendered strokes do not overly overlap. We follow existing works (Liu et al., 2021) to set hyper-parameters $\lambda_p = 8$, $\lambda_r = 10$, and $\lambda_W = 10$. For the adversarial loss weight, we follow (Hu et al., 2023) and set $\lambda_{dis} = 10$. We have conducted experiments to determine the appropriate weight for the confidence regularization loss in Eq. 8 and ultimately set $\lambda_c = 0.1$ as the default value (see Appx. A for details). We use the AdamW optimizer (Loshchilov & Hutter, 2019) with an initial learning rate of 1e-4 and set weight decay to 1e-2. The model is trained for 100,000 iterations using a batch size of 64.

### 4.2 COMPARISON WITH STATE-OF-THE-ART METHODS

**Quantitative Comparison.** We conduct a quantitative comparison between our method and four state-of-the-art oil painting methods: Stylized Neural Painting (Zou et al., 2021) (an optimization-based model), Paint Transformer (Liu et al., 2021) (a feed-forward neural network-based model),

Im2Oil (Tong et al., 2022) (a traditional search-based model), and Compositional Neural Painter (Hu et al., 2023) (a reinforcement learning-based model). Since the main objective of neural painting is to recreate original images, we directly use the pixel loss $\mathcal{L}_{pixel}$ and the perceptual loss $\mathcal{L}_{pcpt}$ (Johnson et al., 2016) as evaluation metrics. $\mathcal{L}_{pixel}$ calculates the mean $L_1$ distance between the rendered images and the target images at the pixel level. $\mathcal{L}_{pcpt}$ is a perceptual metric based on neural network features, which measures the similarity between a target image and a generated image by comparing their differences in high-level feature maps. Lower values of $\mathcal{L}_{pixel}$ and $\mathcal{L}_{pcpt}$ both indicate a better image reconstruction quality. All painting results are produced at a resolution of $512 \times 512$ pixels, with a maximum of 5000 valid strokes applied. Table 1 shows our results on various datasets. It is intriguing to observe that all methods exhibit loss fluctuations across different datasets, indicating a substantial influence of image content complexity on the painting results. For example, our paintings achieve a lower pixel loss and a higher perceptual loss on the FFHQ dataset compared to the Landscapes and Wiki Art datasets. This difference can be attributed to the nature of the images in each dataset. Although plein-air paintings from the Landscapes dataset exhibit complex compositions, they possess less high-level semantic information compared to the high-definition facial images in the FFHQ dataset. Consequently, the plein-air paintings experience higher pixel loss but lower perceptual loss. This also illustrates the necessity of incorporating both pixel and perceptual loss as evaluation metrics, as they capture different aspects of the painting quality. Compositional Neural Painter leverages additional CelebA-HQ (Karras et al., 2018) and ImageNet (Deng et al., 2009) datasets for training, therefore, its pixel matching on the face dataset is slightly lower than ours, whereas our method achieves better perceptual loss. The quantitative results show that our method significantly reduces the pixel metrics and perceptual metrics between the painted canvas and the target image compared to previous approaches.

**Qualitative Comparison.** We compare our method with state-of-the-art methods, as shown in Figure 3. For a fair comparison, we use the same oil painting brushstrokes for all the methods and set the maximum number of valid strokes at around the magnitude of 5000. It can be observed that Stylized Neural Painting struggles with the uniform block-dividing strategy, resulting in obviously inconsistent boundaries. The faces produced by Stylized Neural Painting on FFHQ, possess blurry facial features and exhibit evident grid patterns. In contrast, the faces repainted by our method faithfully preserve facial details while retaining an oil painting style. Paint Transformer tends to generate coarse-grained strokes, neglecting fine details in the images, and it performs poorly in redrawing the edges of images. When confronted with more complex image content and constrained by a limited number of strokes, the output of Im2Oil tends to exhibit a chaotic stroke pattern, because it samples strokes based on the probability density map of the target image, occasionally leading to the loss of essential details. As shown in the first row of Figure 3, Im2Oil incorrectly samples multiple strokes in the sandy area of the image, resulting in a disordered and distorted representation of the sandy region. Conversely, our method, guided by differential images, achieves painting the image with a fitting collection of brushstrokes. Compositional Neural Painter employs real images for training, assigning brushstrokes based on recognized objects. This approach faces challenges with novel images, where inaccuracies in stroke allocation can occur, leading to a misalignment of the visual center when the image is re-drawn. This issue is evident in the third row of Figure 3. Unlike Compositional Neural Painter, our method does not suffer from problems associated with semantic information in images. In summary, our approach effectively mitigates the issue of inconsistent boundary artifacts while simultaneously generating images with a high level of detail. Even when dealing with complex images, our method ensures both superior drawing quality and high brushstroke efficiency.

**User Study.** To further validate the practical significance of our approach, we conduct a Mean Opinion Score (MOS) study to assess user preferences among automatic oil painting methods. We recruit a total of 30 graduate students from diverse disciplines across our university to participate in the MOS test. To minimize potential biases, participants are evenly distributed across five different majors. We launch a questionnaire website through Gradio (gra, 2023). Each questionnaire entails the random selection of 30 image sets, where each

Table 2: The MOS scores and the average inference times for each method. SNP represents Stylized Neural Painting, PT refers to Paint Transformer, and CNP means Compositional Neural Painter. Our approach surpasses the comparison methods in preference score by a clear margin and also offers faster inference speed.

| Method | SNP | PT | Im2Oil | CNP | Ours |
|---|---|---|---|---|---|
| MOS Scores ↑ | 0.25 | 1.26 | 1.87 | 2.12 | 4.47 |
| Inference Time (s) ↓ | 89 | 0.70 | 125 | 12 | 0.72 |

set includes one target image accompanied by five corresponding oil paintings. The identities of the five oil paintings are concealed within each set, and their presentation order is randomized. Participants are tasked with reviewing each set of oil paintings and selecting the two they perceive to be of the highest quality. For each image set, the method associated with each chosen painting receives a score of five points. Furthermore, three specific sets of oil paintings are duplicated in the MOS test. If the selections from these repeated sets vary, this discrepancy raises concerns about the reliability of the participant's judgments. Scores from participants deemed unreliable based on inconsistent selections are excluded. Fortunately, none of the 30 participating volunteers displayed signs of unreliability. Upon completion of the test by all participants, we calculate the average score for each method. The voting results are tabulated in Table 2. Overall, users show a stronger preference for our oil painting compared to other competing paintings. Our method receives a high preference score of 4.47, which is considerably higher than the score of 2.12 earned by the Compositional Neural Painter, which comes in second place.

**Efficiency Analysis.** We measure the average inference times required for each method using a single NVIDIA 3090Ti GPU. For each method, we employ the default settings provided in the official code, and all test images are uniformly sized at $512 \times 512$. The average inference time is reported in Table 2. Due to the streamlined network architecture, our method achieves a significantly higher inference speed compared to Stylized Neural Painting, Im2Oil, and Compositional Neural Painter, with only a 0.02-second difference from Paint Transformer. Nonetheless, the precision of our painting results markedly surpasses that of Paint Transformer.

## 4.3 ABLATION STUDIES AND FURTHER DISCUSSION

**Quantitative Effect of Different Components.** To validate the effectiveness of the key components of our painter framework, we train four ablated models: one variant without the differential image; one variant without the confidence regularization in Eq. 8; one variant without CoordConv layers; and one variant without the WGAN-based discriminator. Table 1 shows the quantitative results. The variant without the differential image exhibits the highest pixel loss, which is 50.0% greater than that of the full model. This validates that incorporating differential images into the model can significantly enhance the accuracy of the paintings. The variant without the CoordConv layers has the highest perceptual loss, indicating that the introduction of positional information is also essential. The full model still outperforms both the variant lacking confidence regularization and the variant without a discriminator, which underscores the necessity of these components.

**Qualitative Effect of Different Components.** The qualitative results are presented in Figure 5. All paintings are produced at a resolution of $256 \times 256$ pixels. It can be seen that the paintings, generated by the variant without differential images as queries, fail to focus on subtle changes in image details and tend to produce coarse strokes. For example, the smooth color gradient in the clouds from the first image in Figure 5 (c) is not well-represented in the painting. The variant without the confidence loss utilizes a greater number of valid strokes to reconstruct the image. The variant without CoordConv layers fails to perceive positional information. As illustrated in Figure 5 (e), it generates many erroneous strokes along the edges of the image. Comparing Figure 5 (f) with Figure 5 (b), the variant without the discriminator, although employing fewer valid strokes, experiences a loss in the fine details of the image.

## 5 CONCLUSION

In this work, we introduce a new automatic oil painting method guided by differential images, which generates brushstrokes akin to those created by human artists. We design a Differential Query Transformer and incorporate the differential image features as queries for decoding the brushstrokes. This "Look, Compare and Draw" approach enables the model to precisely focus on the visual effects produced by the incremental addition of strokes. Coupled with adversarial training, this mechanism significantly improves stroke prediction accuracy and, subsequently, enhances the fidelity of the output images. We have conducted experimental comparisons against state-of-the-art stroke-based painting methods on unseen real-world datasets and validated the superiority of our method through a combination of qualitative and quantitative evaluations, as well as a user study, assessing both pixel-level and perception-level reconstruction accuracy.

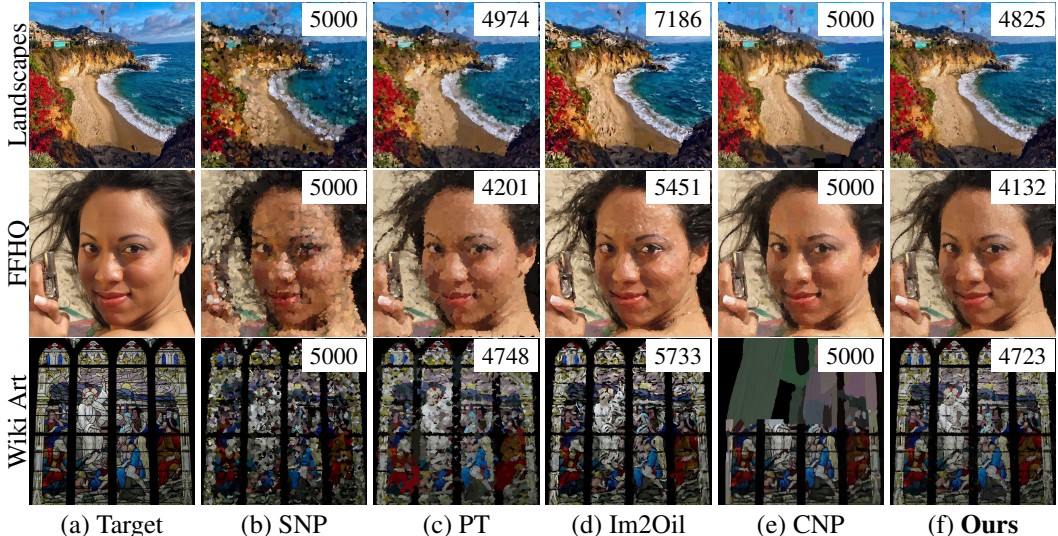

Figure 3: Qualitative comparison between our model and state-of-the-art neural painting methods on unseen real-world datasets.The maximum number of valid strokes is set to 5000 for each model. We set the sampling rate for Im2Oil to 1/9. The actual number of brushstrokes used in the painting is annotated in the top right corner of the image. We observe that the proposed method shows better visual quality using relatively fewer strokes. Please zoom in to obtain a more detailed view.

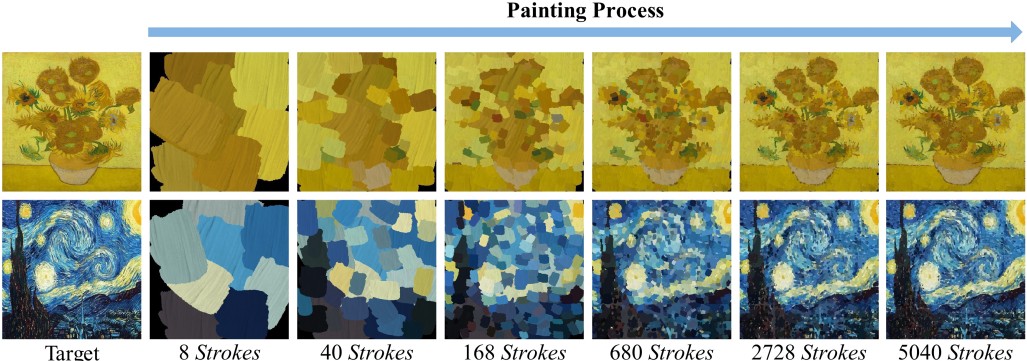

Figure 4: Our painting progress following a coarse-to-fine manner.

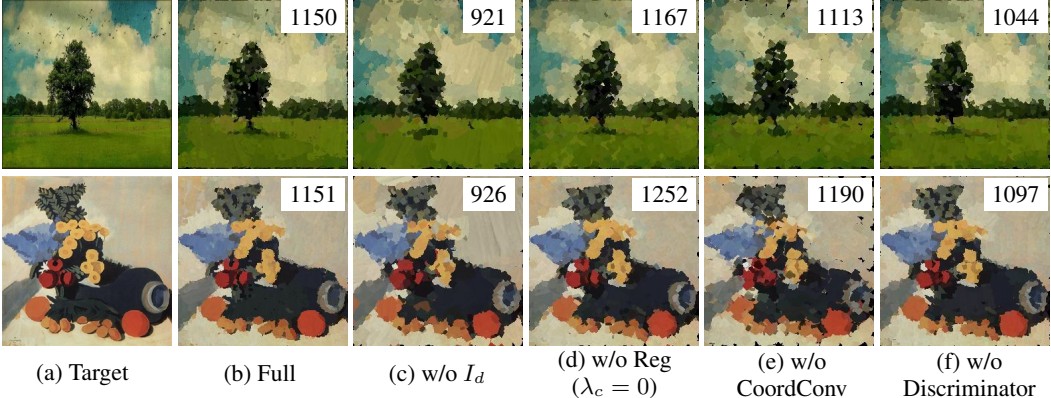

Figure 5: Ablation study on the primary components of our framework. All painting results are produced at a resolution of $256 \times 256$ pixels. The actual number of brushstrokes used in the painting is annotated in the top right corner of the image.

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

## A  EFFECT OF THE WEIGHT $\lambda_c$

We investigate the influence of varying weights ($\lambda_c$) for the confidence regularization on model performance. As shown in Table 3, we observe that when $\lambda_c > 1$, both the pixel loss and perceptual loss of the model are relatively high, indicating poor image quality. When $\lambda_c < 0.5$, the model exhibits relatively lower pixel loss, and when $\lambda_c = 0.1$, the model achieves the minimum perceptual loss. Consequently, based on the experimental results, we set $\lambda_c = 0.1$ as the default value.

Table 3: Ablation study on the weight $\lambda_c$. We set $\lambda_c = 0.1$ as the default value.

| $\lambda_c$ | 0.05 | 0.1 | 0.2 | 0.5 | 1 | 5 | 10 |
|---|---|---|---|---|---|---|---|
| $\mathcal{L}_{pixel} \downarrow$ | 0.048 | 0.046 | 0.046 | 0.050 | 0.050 | 0.055 | 0.058 |
| $\mathcal{L}_{pcpt} \downarrow$ | 0.668 | 0.607 | 0.614 | 0.686 | 0.685 | 0.786 | 0.791 |

## B  PAINTING INFERENCE ALGORITHM

---
**Algorithm 1** Painting Inference Algorithm

---
**Input:** a target image $I_t$ with shape $H \times W$; Patch size P;
**Output:** a rendered image $\hat{I}_t$ and ordered strokes $\hat{S}_t$;
  1: #: Calculate the scale number.
  2: $K = \max\left(\operatorname{argmin}_K \left\{ P \times 2^K \geqslant \max(H, W) \right\}, 0\right)$;
  3: $I_c = blank\_canvas$;
  4: $\hat{S}_t = \varnothing$
  5: #: Iteration among different scales.
  6: **for** $0 \leqslant k \leqslant K$ **do**
  7:    Resize $I_t$ and $I_c$ to a size of $(P \times 2^k, P \times 2^k)$;
  8:    The differential image $I_d = I_t - I_c$.
  9:    Divide $I_t$, $I_c$ and $I_d$ uniformly into multiple patches of size $(P, P)$;
10:    Given the two corresponding patches from $I_t$ and $I_c$ and the differential patches in $I_d$, Local Encoder and DQ-Transformer predict the stroke sets for each location. We aggregate all patch strokes as $\left(\hat{S}_t^k, \hat{C}_t^k\right)$;
11:    #: Here we only draw high-confidence strokes.
12:    $I_c = I_c + renderer\left(I_c, \hat{S}_t^k, \hat{C}_t^k\right)$;
13:    $\hat{S}_t = \hat{S}_t \cup selected(\hat{S}_t^k)$
14: **end for**
15: Pad $I_t$ and $I_c$ to a size of $(P \times 2^K + P, P \times 2^K + P)$;
16: #: Make up the boundary areas.
17: Predict and render the stroke sets $\left(\hat{S}_t^{K+1}, \hat{C}_t^{K+1}\right)$ onto the extended $I_c$;
18: $\hat{S}_t = \hat{S}_t \cup selected(\hat{S}_t^{K+1})$
19: $\hat{I}_t = crop(I_c, size = (H, W))$;
20: **Return** $\hat{I}_t$ and $\hat{S}_t$.

---

