# OpenReview forum: "Look, Compare and Draw:  Differential Query Transformer for Automatic Oil Painting"
_ICLR.cc/2025/Conference — ICLR 2025 Conference Withdrawn Submission_

### Official Review · Reviewer_qEMm · 2024-10-20

**Soundness:** 2
**Presentation:** 3
**Contribution:** 1
**Rating:** 1
**Confidence:** 5

**Summary:**

This paper introduces an automatic oil painting method which utilized a differential image as a guidance condition, inspired by the “Look, Compare and Draw” idea. A differential Query Transformer is designed to incorporate the differential image feature as queries. Experiments are conducted between several state-of-the-art methods to show its performance.

However, both the method design and the writing of this paper are highly similar to the previous paper PaintTransformer (ICCV 2021), detailed in "Weaknesses", with limited technical novelty, which makes this paper hardly satisfy the standards of ICLR.

**Strengths:**

1. This paper proposes an automatic oil painting which uses the differential image as an extra condition information input, and the differential image features as queries.
2. The quantitative results are slightly better than previous methods.

**Weaknesses:**

1. The technical novelty is somewhat limited.
The whole pipeline is quite similar to PaintTransformer [1]. The main difference to PaintTransformer is using the features of a differential image (between the current canvas and target image) as queries to the Transformer, to predict stroke parameters. Except for this design, the other parts are very similar to PaintTransformer (Detailed below).

The entire Methodology Section (Sec. 3) is quite similar to PaintTransformer, including the method design and the writing.

a) The self-supervised pipeline where the strokes are randomly synthesized in Sec. 3.1, is the same as the self-training pipeline in PaintTransformer[1] Sec. 3.1, but without any citation.

PaintTransformer[1]: “To train a Stroke Predictor, as shown in Fig. 2, we propose a novel self-training pipeline which utilizes randomly synthesized strokes.”

This paper: “We employ a self-supervised pipeline in which the current canvas and target images are constructed using randomly synthesized strokes, eliminating the need for real images during training”.

b) The stroke renderer in Sec. 3.2 is the same as previous methods, e.g., PaintTransformer[1], and CNP [2]. Moreover, the writing is highly similar.

CNP[2] Sec 3.4: “The strokes parameters are 𝑠 = {𝑥,𝑦,𝑤,h,𝜃,𝑟,𝑔,𝑏}, where (𝑥, 𝑦) indicate the coordinate of the stroke center, 𝑤, h are the width and height of the stroke, 𝜃 is the rotation angle and (𝑟,𝑔,𝑏) is the RGB color of the stroke.”

This paper Sec 3.2: “The strokes parameters are s = {x, y, h, w, θ, r, g, b}, where (x, y) denotes the coordinates of the stroke center, h, w represent the height and width of the stroke, θ is the rotation angle, and (r, g, b) indicates the RGB values of the stroke.”

c) The training loss in Sec. 3.4 are the simple combination of PaintTranformer[1] and CNP[2], but without any citation. Moreover, even the formulas are almost the same as previous papers. The loss terms in Eq. (3)(4)(5)(6)(7) are exactly the same as the loss terms in PaintTransformer[1] Eq. (7)(8)(10)(11)(13). The adversarial loss part (Eq. (9)), introducing WGAN-GP into the training, is the same as CNP[2] Eq. (5).

d) The inference pipeline in Sec. 3.5 is the same as PaintTransformer[1] Sec. 3.5, but without any citation. Again, writing is highly similar to PaintTransformer[1].

PaintTransformer[1] Sec 3.5: “To imitate a human painter, we devise a coarse-to-fine algorithm to generate painting results during inference…Target image and current canvas would be cut into several non-overlapping P × P patches before being sent to Stroke Predictor.”

This paper Sec 3.5: “To generate painting strokes that mimic human artists, we predict strokes in a coarse-to-fine manner during the inference process… Both the target image and the current canvas are uniformly divided into multiple patches with a size of P × P, which are then fed into our painting network for stroke prediction.”

e) The Sec. 3.3 is the only part that is slightly different from previous methods.

If we compare the Sec. 3.3 of this paper to Sec. 3.3 of PaintTransformer, this paper and PaintTransformer[1] both use a Transformer structure for stroke prediction. Both methods extract features for canvas and target image, use a Transformer encoder and a Transformer decoder, and use two MLPs to predict stroke parameters and stroke confidences.

The main difference is that, instead of only extracting features $F_c$, $F_t$ for the canvas $I_c$ and target image $I_t$, this paper additionally extracts feature $F_d$ for the differential image $I_d=I_t-I_c$, and the $F_d$ is transformed into query tokens for the Transformer Decoder, while in PaintTransformer, the query for Transformer Decoder is learnable. This difference is a slight change to the inputs of the Transformer.


2. The quantitative and qualitative comparison are not comprehensive. The authors only compare under the setting of 5000 strokes. However, it is necessary to test under different number of strokes. Other strokes number setting should also be done, e.g., 200, 500, 1000, 3000 strokes, like CNP [2].

3. From the qualitative comparisons in Fig. 3, the differences between the proposed method and other methods are not quite visible. However, in the results of MOS in the Tab. 2, the gap of MOS between different methods is quite big. The MOS results are inconsistent with the qualitative comparisons. It is suggested to explain this inconsistency, e.g., give the detailed information of the user study.

4. From the qualitative ablation study results in Fig.5, it is difficult to see the difference, especially the Fig. 5(f).

[1] Liu S, Lin T, He D, et al. Paint transformer: Feed forward neural painting with stroke prediction[C]//Proceedings of the IEEE/CVF international conference on computer vision. 2021: 6598-6607.

[2] Hu T, Yi R, Zhu H, et al. Stroke-based neural painting and stylization with dynamically predicted painting region[C]//Proceedings of the 31st ACM International Conference on Multimedia. 2023: 7470-7480.

**Questions:**

Please refer to the questions in the “Weaknesses” part.

---

> ### Author Response · Authors · 2024-11-14
>
> Dear Reviewer,
> Thank you for your thoughtful feedback on our paper. We appreciate your detailed analysis and constructive suggestions.
>
> Q1. The whole pipeline is quite similar to PaintTransformer.
> A: Yes, we follow the settings from PaintTransformer, so some of the task configurations are indeed the same. We will give more credit and citations to PaintTransformer. We also want to emphasize that our three contributions are distinct from those of PaintTransformer, and our results are notably better, as evidenced by both quantitative and qualitative comparisons. Additionally, the loss component is not one of our contributions, but we include it for the completeness of our work. We will reduce the proportion of this part and highlight our contributions instead.
> The pixel loss and perceptual loss of our model and PaintTransformer when painting images at different scales are shown in the table below:
> | size      | 128*128          | 128*128          | 256*256          | 256*256          | 1024*1024        | 1024*1024        |
> | --------- | ---------------- | ---------------- | ---------------- | ---------------- | ---------------- | ---------------- |
> | loss      | pixel            | perceptual       | pixel            | perceptual       | pixel            | perceptual       |
> | PT        | 0.086            | 0.865            | 0.072            | 0.946            | 0.058            | 0.775            |
> | Ours      | 0.067            | 0.797            | 0.057            | 0.815            | 0.042            | 0.535            |
>
> Our pixel loss and perceptual loss are significantly lower than those of PaintTransformer.
> We will revise the organization of our paper to focus more on the differential guidance part and make the loss section more concise.
>
> Q2. Other strokes number setting.
> A: Among the four methods we compare, only SNP and CNP can set the exact number of strokes. PaintTransformer and Im2Oil can only roughly control the number of strokes by adjusting the setting parameters. We selected 5000 strokes, aiming for a relatively fair stroke quantity for each method.
>
> Q3. The MOS results.
> A: In the MOS study, we ask each participant to select the **two** images they consider to have the best painting effect out of five. Our method produces consistent and high-quality painting results across various datasets, giving it a higher probability of being selected by the participants. Im2Oil and CNP also show good painting quality, but their performance is less stable across different images, leading to slightly lower scores than ours. SNP and PT lack the ability to depict details compared to other methods, resulting in a lower score.
>
> Q4. Qualitative ablation study results in Fig.5.
> A: When zooming in on the images, the result of （b）in Fig.5 appears smoother than that of （f）. We will provide a more detailed discussion in the subsequent sections of the paper.

---

### Official Review · Reviewer_7DCJ · 2024-11-03

**Soundness:** 3
**Presentation:** 4
**Contribution:** 3
**Rating:** 6
**Confidence:** 4

**Summary:**

The paper studies the popular problem of automated painting generation given a reference input image. Unlike image stylization methods, the paper focusses on the stroke-based methods which consider the generation of the final painting through a sequence of strokes, which when applied to a canvas lead to output painting similar to the input image. Prior works in this area typically adopt and reinforcement learning or stroke optimization based methods which struggle with 1) repeated or duplicated stroke patterns, and 2) lack of human-like efficiency during the painting process. To address these problems and incorporate a more human-like inductive bias into the generation process this paper proposes the idea of look, compare and draw. In particular, the core idea boils down to using the differential image features (between the current canvas and the target image), which are used as queries (after projection layer) to the DQ-transformer network in order to decode the stroke parameters. Experiments on three different datasets are provided in order to demonstrate the stroke-efficiency of the proposed approach.

**Strengths:**

* The paper studies an interesting problem of using stroke-based synthesis for automated oil painting.
* In particular, it proposes an interesting direction towards adding an human-like inductive bias into the painting process.
      - While the problem of using a human-like painting process for efficiency has been studied before (Intelli-paint), the paper introduces the use of differential image features, which are as used as queries in the transformer attention layers for stroke decoding.
      - The differential image features (Fig. 1), provide a useful cue on which image features the model should focus on next, and therefore provides good inductive-bias for improving output quality while using less number of brush-strokes.
* Both qualitative and quantitive evaluations are provided in order to compare the proposed approach with prior works.

**Weaknesses:**

* One of my concerns is the use of synthetic data and self-supervised pipeline (similar to paint-transformer paper) while encoding a human-like inductive bias into the painting process.
     - For instance, while the use of differential features in Fig. 1, point to the semantically meaningful regions during the painting process, the same is unclear when using a synthetic data and random brushstrokes when adopting a self-supervised training approach.
    - Do the authors have any intuition on the above?

*  The problem of stroke-efficiency has also been studied in [1]. In particular, it proposes a simple gradient-optimization of the original stroke parameters in order to improve the stroke efficiency in a post-hoc manner.
   - While the original paper uses Bezier curve stroke representation and the current paper uses oil painting strokes, it would be interesting to see if the brushstroke regularization from [1] can be used to further improve stroke-efficiency of the painting process (Fig. 3).

* While a minor point, does the proposed approach generalize to other stroke representations as in SNP paper [2]?


References:
[1] Intelli-Paint: Towards Developing Human-like Painting Agents, Singh et al., ECCV 2022
[2] Stylized neural painting, Zou et al, 2021

**Questions:**

Please see weakness section above.

---

> ### Author Response · Authors · 2024-11-14
>
> Dear Reviewer,
> Thank you for your thoughtful feedback on our paper. We appreciate your detailed analysis and constructive suggestions.
>
> Q1. The use of differential features in self-supervised pipeline.
> A: We introduce differential features with the aim of directing the model to pay more attention to the areas where there are differences between the canvas and the target, focusing more on strokes in regions with larger discrepancies. These differential features do not contain semantic information. The model does not need to consider semantic information during both the training and inference stages.
>
> Q2. The brushstroke regularization from **Intelli-Paint**.
> A: Intelli-Paint introduces importance vectors to evaluate the significance of each brushstroke action. This appears to serve the same function as the stroke confidence in our method.
>
> Q3. Generalization to other stroke representations as in the **SNP** paper.
> A: The method we propose can be extended to other stroke representations mentioned in the SNP paper, requiring only adjustments to the stroke parameters.
>
> All the papers mentioned above will be cited and discussed in our manuscript.

---

### Official Review · Reviewer_5ZtB · 2024-11-08

**Soundness:** 3
**Presentation:** 3
**Contribution:** 2
**Rating:** 6
**Confidence:** 4

**Summary:**

This paper proposes a novel automatic oil painting method called the Differential Query Transformer (DQ-Transformer). By incorporating differential image analysis into the neural network, the model can effectively focus on the incremental impact of successive brushstrokes, generating more dynamic and expressive strokes. The method was evaluated both quantitatively and qualitatively on three public datasets, demonstrating superior performance in pixel-level and perception-level reconstruction, and earning higher user preference. The DQ-Transformer consists of two main components: the Local Encoder, which extracts image features, and the DQ-Transformer, which generates stroke parameters. The training objectives include pixel loss, stroke loss, and adversarial loss, ensuring that the generated images resemble the target image and possess authentic oil painting textures. Experimental results show that this method achieves faster inference speed while maintaining high accuracy.

**Strengths:**

1. The paper is very well written and the discussion of related work covers most literatures I know.
2. The paper conduct quantitative and qualitative experiments on three public datasets (Landscapes, FFHQ, and Wiki Art), demonstrating that the proposed method achieves better results in both pixel-level and perception-level reconstruction. It also receive higher user feedbacks across various painting themes. Additionally, the method excels in stroke efficiency, achieving competitive painting quality with fewer strokes.
3. Based on Figure 3, it can be observed that the level of detail depicted by this method has reached a new standard.

**Weaknesses:**

1. This paper indeed makes some beneficial improvements to the Paint transformer, but for me, the paper is too focused on craftsmanship.
The main goal of the paper is to reproduce a painting as closely as possible, as shown in Figure 3. Although previous work has some flaws, it is difficult for this paper to make a significant difference in this dimension.

2. Some sections of this paper, such as the Adversarial Loss, seem to describe content that does not differ significantly from previous work in this field. This part should be appropriately simplified because it is not closely related to the main contributions of this paper.

**Questions:**

This question is not entirely directed at this paper; it is a common issue within this field. Namely, why do we use this form to study AI painting and pursue extreme effects? I believe a recent paper from Siggraph Asia (I'm not sure if it has been published yet) will provide some insights: **ProcessPainter: Learn Painting Process from Sequence Data**.

Another interesting related work is **Clipdraw: Exploring text-to-drawing synthesis through language-image encoders**. If the method proposed in this paper can divide the painting process into different stages, can we pursue different painting goals at each stage?

I believe that our efforts to enhance AI's proficiency with painting tools should ultimately be integrated with painting education and even the creation of art.

Overall, I would like the authors to state in the rebuttal phase what content they intend to remove and add in the revised version of the paper.

---

> ### Author Response · Authors · 2024-11-14
>
> Dear Reviewer,
> Thank you for your thoughtful feedback on our paper. We appreciate your detailed analysis and constructive suggestions.
>
> As mentioned in **ProcessPainter**, an automatic oil painting method based on strokes can provide a large dataset of painting processes from coarse to fine for techniques based on diffusion models. Our method features fast inference speed and delivers good painting results, making it capable of providing training samples with painting processes for AI at a relatively low cost.
>
> **CLIPDraw** achieves different stages of drawing effects through various textual descriptions, making it a useful visual application tool for assisting AI art. Moreover, from our perspective, the contribution of our stroke-based automated oil painting technique to the field of AI art research lies in its ability to provide large-scale AI models with stable and high-quality datasets.
>
> All the papers mentioned above will be cited and discussed in our manuscript. We will revise the organization of our paper to focus more on the differential guidance part and make the loss section more concise.

---

### Note · Authors · 2024-11-15

I have read and agree with the venue's withdrawal policy on behalf of myself and my co-authors.